# Sitafloxacin for Third-Line *Helicobacter pylori* Eradication: A Systematic Review

**DOI:** 10.3390/jcm10122722

**Published:** 2021-06-20

**Authors:** Toshihiro Nishizawa, Munkhbayar Munkjargal, Hirotoshi Ebinuma, Osamu Toyoshima, Hidekazu Suzuki

**Affiliations:** 1Department of Gastroenterology and Hepatology, International University of Health and Welfare, Narita Hospital, Narita 286-8520, Japan; nisizawa@kf7.so-net.ne.jp (T.N.); mb99md@gmail.com (M.M.); ebinuma@me.com (H.E.); 2Gastroenterology, Toyoshima Endoscopy Clinic, Tokyo 157-0066, Japan; t@ichou.com; 3Division of Gastroenterology and Hepatology, Department of Internal Medicine, Tokai University School of Medicine, Isehara 259-1193, Japan

**Keywords:** sitafloxacin, vonoprazan, *H. pylori*

## Abstract

Background and Aim: Sitafloxacin-based therapy is a potent candidate for third-line *Helicobacter pylori* eradication treatment. In this systematic review, we summarise current reports with sitafloxacin-based therapy as a third-line treatment. Methods: Clinical studies were systematically searched using PubMed, Cochrane library, Web of Science, and the Igaku-Chuo-Zasshi database. We combined data from clinical studies using a random-effects model and calculated pooled event rates, 95% confidence intervals (CIs), and the pooled odds ratio (OR). Results: We included twelve clinical studies in the present systematic review. The mean eradication rate for 7-day regimens of either PPI (proton pump inhibitor) or vonoprazan–sitafloxacin–amoxicillin was 80.6% (95% CI, 75.2–85.0). The vonoprazan–sitafloxacin–amoxicillin regimen was significantly superior to the PPI–sitafloxacin–amoxicillin regimen (pooled OR of successful eradication: 6.00; 95% CI: 2.25–15.98, *p* < 0.001). The PPI–sitafloxacin–amoxicillin regimen was comparable with PPI–sitafloxacin–metronidazole regimens (pooled OR: 1.06; 95% CI: 0.55–2.07, *p* = 0.86). Conclusions: Although the 7-day regimen composed of vonoprazan, sitafloxacin, and amoxicillin is a good option as the third-line *Helicobacter pylori* eradication treatment in Japan, the extension of treatment duration should be considered to further improve the eradication rate. Considering the safety concern of fluoroquinolones, sitafloxcin should be used after confirming drug susceptibility.

## 1. Introduction

*Helicobacter pylori* (*H. pylori*) is associated with gastric cancer, mucosa-associated lymphoid tissue lymphoma, atrophic gastritis, and peptic ulcer [1,2,3]. Accordingly, *H. pylori* eradication can effectively prevent and treat these diseases [4,5]. However, a recent increase in antibiotic resistance has complicated *H. pylori* eradication [6,7,8,9]. Some patients necessitate the use of third-line treatment [10]. Available options for third-line treatment include fluoroquinolones and rifabutin [11,12,13]. In terms of fluoroquinolones, sitafloxacin has demonstrated excellent in vitro efficacy against *H. pylori* when compared with levofloxacin and garenoxacin [14,15]. Murakami et al. compared sitafloxacin-based therapy, levofloxacin-based therapy, and high dose proton pump inhibitor (PPI)–amoxicillin therapy in a randomised controlled trial (RCT) [16] and observed superior results with sitafloxacin-based therapy. In 2016, a Japanese guideline recommended sitafloxacin-based triple therapy instead of levofloxacin-based triple therapy [17]. Several studies have evaluated sitafloxacin-based triple therapy as third-line treatment [18,19,20,21,22,23,24,25,26,27]. A novel potassium-competitive acid blocker (vonoprazan), deemed superior to PPIs, has also been introduced [28,29]. Several variations of sitafloxacin-based triple therapy in terms of combinations, doses, and treatment durations have been employed. In the present systematic review, we summarise up-to-date reports with sitafloxacin-based therapy as third-line treatment.

## 2. Methods

### 2.1. Search Strategy

Relevant reports were systematically searched using PubMed, Cochrane library, Web of Science, and Igaku-Chuo-Zasshi database in Japan (up to May 2021) [30]. The search words used were as follows: (sitafloxacin OR DU-6859a OR Gracevit) AND (*Helicobacter pylori* OR *H. pylori* OR *Helicobacter* infection) AND (eradication OR treatment OR therapy). There was no language limitation applied.

### 2.2. Inclusion and Exclusion Criteria

The inclusion criteria were as follows: (1) study design: clinical study; (2) participants: patients who failed first-line and second-line eradication treatments; (3) intervention: third-line eradication treatment using sitafloxacin; (4) outcome: confirmation of *H. pylori* eradication at least 4 weeks after therapy. The exclusion criteria were as follows: (1) meeting abstracts; (2) review articles; (3) duplication.

### 2.3. Data Extraction

The following data were extracted from all included studies: principal author, publishing year, study type, drugs and doses for *H. pylori* eradication regimens, duration of treatment, the number of enrolled subjects, diagnostic methods employed for testing *H. pylori* infection after eradication, and key outcome data such as eradication rates, as well as the occurrence of diarrhoea and severe adverse events. The eligibility of all articles was independently examined by two reviewers (T.N. and M.M.). Disagreements were resolved by consulting a third reviewer (O.T.).

### 2.4. Assessment of Methodological Quality

We evaluated the quality of enrolled studies using the Jadad scale [31] for RCTs or by the Quality Assessment Tool for Case Series Studies (QATCSS) of the National Institutes of Health for nonrandomised, open label studies (National Heart, Lung, and Blood Institute. Available from: https://www.nhlbi.nih.gov/health-pro/guidelines/in-develop/cardiovascular-risk-reduction/tools/case_series, accessed on 1 May 2021) for nonrandomised studies. Two reviewers (T.N. and H.E.) assessed the quality of the included studies.

### 2.5. Statistical Analysis

Statistical analysis was performed using Comprehensive Meta-Analysis (CMA) software (version 3, Biostat, Inc., Englewood, NJ, USA) and Review Manager (RevMan; The Cochrane Collaboration, 2008; The Nordic Cochrane Centre, Copenhagen, Denmark) [32]. Pooled event rates and 95% confidence intervals (CIs) were calculated using a random-effects model. The odds ratio (OR) for successful eradication was calculated using the random-effects model and the Mantel–Haenszel method [33]. Heterogeneity between studies was assessed by Cochran’s Q and I^2^ tests [7]. *p*-value < 0.1 was considered as significant heterogeneity, as the power of the Q test is low. An I^2^ score ≥ 50% was considered a moderate level of heterogeneity [34].

## 3. Results

### 3.1. Search Results

The systematic review process identified 248 potential reports (Figure 1). Based on the exclusion criteria, we then excluded 230 studies (32 duplications, 25 unrelated topics, 63 review articles, 18 protocols for clinical trials, 90 meeting abstracts, and 2 case reports). The remaining 18 studies were scrutinised, and six additional studies were rejected. In four studies, the sitafloxacin-containing regimen was used as first-line or second-line treatment for subjects with penicillin allergy [35,36,37,38]. One study combined the eradication rates of levofloxacin-based and sitafloxacin-based therapy [39]. One study combined the eradication rates of first, second, and third-line therapy [9]. Finally, we included 12 clinical studies in the present systematic review [16,18,19,20,21,22,23,24,25,26,27,40]. The included studies were all performed in Japan. In Sugimoto et al.’s, Saito et al.’s and Sue et al.’s studies, the first-line therapy was the 7-day regimen with vonoprazan or PPI, clarithromycin 200 mg or 400 mg b.i.d., and amoxicillin 750 mg b.i.d., and the second-line therapy was the 7-day regimen with vonoprazan or PPI, metronidazole 250 mg b.i.d., and amoxicillin 750 mg b.i.d. In another eight studies, the first-line therapy was the 7-day regimen with PPI, clarithromycin, and amoxicillin, and the second-line therapy was the 7-day regimen with PPI, metronidazole, and amoxicillin. In all included studies, the sitafloxacin dose was 100 mg b.i.d.

### 3.2. Efficacy of PPI–Sitafloxacin–Amoxicillin

Table 1 summarises the efficacy of PPI–sitafloxacin–amoxicillin, which is sorted in the order of treatment duration and amoxicillin dose employed. The 1500 mg/day amoxicillin dose was administered as 750 mg b.i.d., while the 2000 mg/day was administered as 500 mg q.i.d. Prolonged treatment durations appeared to improve eradication rates. However, in an RCT comparing 7-day and 14-days regimens, Furuta et al. found no significant difference [23]. Furthermore, Mori et al. compared 7-day and 10-days regimens and observed was no significant difference [25].

Three studies evaluated the efficacy of the vonoprazan–sitafloxacin–amoxicillin regimen (Table 2) and reported eradication rates ranging between 83.3% and 93.0% in the per-protocol analysis.

Figure 2 shows the forest plot of eradication rates for 7-day regimens of either PPI or vonoprazan–sitafloxacin–amoxicillin. The mean eradication rate for 7-day regimens of either PPI or vonoprazan–sitafloxacin–amoxicillin was 80.6% (95% CI, 75.2–85.0). In the subgroup analyses, the mean eradication rates were 70.1% (95% CI, 59.0–79.2%) for PPI–sitafloxacin–amoxicillin 1500 mg/day, 84.4% (95% CI, 76.7–90.0%) for PPI–sitafloxacin–amoxicillin 2000 mg/day, 88.9% (95% CI, 75.5–95.4%) for vonoprazan–sitafloxacin–amoxicillin 1500 mg/day, and 87.5% (95% CI, 73.3–94.7%) vonoprazan–sitafloxacin–amoxicillin 2000 mg/day.

Two clinical studies compared PPI–sitafloxacin–amoxicillin and vonoprazan–sitafloxacin–amoxicillin regimens [20,22] and revealed that the vonoprazan–sitafloxacin–amoxicillin regimen was significantly superior to the PPI–sitafloxacin–amoxicillin regimen in the per-protocol analysis (pooled OR of successful eradication 6.00; 95% CI: 2.25–15.98, *p* < 0.001) (Figure 3A). The intention-to-treat analysis had similar results (Figure 3B).

### 3.3. Efficacy of PPI–Sitafloxacin–Metronidazole

Several studies evaluated the efficacy of the PPI–sitafloxacin–metronidazole regimen. Table 3 summarises these results and is organised in the order of treatment duration. Furuta et al. compared 7-day and 14-days regimens in an RCT and observed no significant difference [23].

### 3.4. PPI–Amoxicillin–Sitafloxacin versus PPI–Metronidazole–Sitafloxacin

Three sets of RCTs compared PPI–sitafloxacin–amoxicillin and PPI–sitafloxacin–metronidazole regimens [23,24]. In each of these studies, the amoxicillin dose employed was 2000 mg/day (500 mg q.i.d.), and the metronidazole dose was 500 mg/day (250 mg b.i.d.). The treatment durations were 7, 10, and 14, respectively. Owing to the limited number of studies, RCTs that employed different criteria were included in this meta-analysis. No significant difference in eradication rates was observed between PPI–sitafloxacin–amoxicillin and PPI–sitafloxacin–metronidazole regimens in the per-protocol analysis (pooled OR: 1.06, 95% CI: 0.55–2.07, *p* = 0.86, Figure 4A). Additionally, no heterogeneity was detected between RCTs (*p* = 0.60, I^2^ = 0%). The intention-to-treat analysis had similar results (Figure 4B).

### 3.5. Adverse Events

Figure 5 shows the forest plot of adverse event (diarrhoea) associated with 7-day regimens composed of PPIs or vonoprazan–sitafloxacin–amoxicillin. The pooled adverse event (diarrhoea) rate was 24.4% (95% CI: 16.7–34.3%). The frequency of diarrhoea ranged between 12.5% and 50.0%. Sue et al. reported two cases (6.1%) presenting severe allergy. Other studies did not report any severe adverse events.

### 3.6. Quality Assessment

Quality assessment is reported in Table 4. A maximum score of 5 was obtained using the Jadad scale, whereas a maximum score of 9 was attained with QATCSS. In general, the quality of the included studies was good, apart from the study by Tokunaga et al.

## 4. Discussion

The 7-day regimen composed of vonoprazan, sitafloxacin, and amoxicillin is considered as a good option for the third-line *H. pylori* eradication treatment in Japan.

An in vitro study has reported that sitafloxacin and metronidazole exhibit a synergistic antimicrobial activity, while sitafloxacin and amoxicillin failed to demonstrate such activity [41]. Several RCTs have compared PPI–sitafloxacin–metronidazole with PPI–sitafloxacin–amoxicillin [23,24]. This meta-analysis confirmed that the efficacy of PPI–sitafloxacin–amoxicillin was comparable with that of PPI–sitafloxacin–metronidazole. The PPI–sitafloxacin–metronidazole regimen is a good option in patients presenting penicillin allergy [36,38].

In PPI–sitafloxacin–amoxicillin, amoxicillin is administered at quantities of 1500 mg/day (750 mg b.i.d.), or 2000 mg/day (500 mg q.i.d.). The bactericidal effect of amoxicillin depends on the time above the minimum inhibitory concentration (MIC), as amoxicillin has minimal post-antibiotic effect [42,43]. Furuta et al. compared amoxicillin 750 mg b.i.d., 500 mg t.i.d., and 500 mg q.i.d. in a PPI–metronidazole–amoxicillin regimen, and reported eradication rates of 80.5%, 90.5%, and 95.2%, respectively, indicating that four times daily amoxicillin dosing maximised the eradication rate [44]. This study indicated that the optimal amoxicillin dose was 2000 mg/day (500 mg q.i.d.). The amoxicillin dose in the vonoprazan–sitafloxacin–amoxicillin regimen needs to be further investigated in future studies.

Regarding the optimal duration for *H. pylori* eradication therapy, a meta-analysis showed that increasing the duration of PPI-based triple therapy increased the eradication rates [45]. There are many reports that the eradication rates of fluoroquinolone-based therapy were increased by extending the duration to 14 days [46,47,48,49,50,51,52]. Conversely, Furura et al. compared 7-day and 14-days regimens of PPI–sitafloxacin–amoxicillin in their RCT, and found no significant difference. Mori et al. also compared 7-day and 10-days regimens of PPI–sitafloxacin–amoxicillin, and found no significant difference. However, the number of patients was limited, and hence these studies might be underpowered. As for the vonoprazan–sitafloxacin–amoxicillin regimen, there is no data other than 7-days regimen. The 14-day regimen with vonoprazan–sitafloxacin–amoxicillin may achieve excellent eradication rates. The optimal duration of the vonoprazan–sitafloxacin–amoxicillin regimen needs to be further investigated in future studies.

In Japan, the primary resistance of *H. pylori* to levofloxacin is 15% [1]. In contrast, the sitafloxacin resistance rates reported in the included studies ranged between 21.7% and 60.3%; these resistance rates were relatively high. This could be attributed to the fact that patients who failed to demonstrate successful eradication twice might have a history of prior antibiotic use, including fluoroquinolones for non-eradication purposes [53,54]. Fluoroquinolone resistance is caused by *gyrA* mutation [55,56,57,58]. Three studies reported both sitafloxacin resistance rates and *gyrA* mutation rates. Accordingly, the sitafloxacin resistance rates and *gyrA* mutation rates were 42.5% and 55.1% in Matsuzaki et al.’s study [21], 58.7% and 60.3% in Mori et al.’s 2016 study [24], and 50.0% and 50.0% in Mori et al.’s 2020 study [25]. When the sitafloxacin resistance was defined as MIC, the sitafloxacin resistance rate was found to be similar to the *gyrA* mutation rate. Eradication rates for *gyrA* mutation-negative and mutation-positive groups were 96.7% and 74.4% in Matsuzaki et al.’s study [21], 100% and 70.3% in Mori et al.’s 2016 study [24], and 89.5% and 68.4% in Mori et al.’s 2020 study, respectively [25]. Sitafloxacin-based triple therapy showed excellent eradication rates for *gyrA* mutation-negative *H. pylori*. Sitafloxacin-based therapy should be used for sitafloxacin susceptible or *gyrA* mutation-negative *H. pylori.* Sitafloxacin resistant or *gyrA* mutation-positive *H. pylori* would require alternate treatments such as rifabutin-based therapy [13,59,60,61].

The United States Food and Drug Administration recently enhanced warnings about disabling and potentially permanent side effects involving the tendons, muscles, joints, nerves, and central nervous system [62]. Balancing risk and benefit should be considered.

Our systematic review has several limitations. The included studies were all from Japan, as sitafloxacin is only available in Japan and Thailand. The availability of vonoprazan is also limited. In Japan, the first-line regimen is PPI–clarithromycin–amoxicillin and the second-line regimen is PPI–metronidazole–amoxicillin. Therefore, the results of the current study may not directly applicable to the populations who were administered bismuth containing regimens as first-line or second-line regimens. Therefore, these results may not be generalisable to other ethnicities and countries.

In conclusion, the 7-day regimen with vonoprazan 20 mg b.i.d., sitafloxacin 100 mg b.i.d., and amoxicillin 750 mg b.i.d. or 500 mg q.i.d. is a good option as the third-line *H. pylori* eradication treatment, at least in Japan. However, the extension of treatment duration should be considered to further improve the eradication rate. Considering the safety concern of fluoroquinolones, sitafloxcin should be used after confirming drug susceptibility.

## Figures and Tables

**Figure 1 jcm-10-02722-f001:**
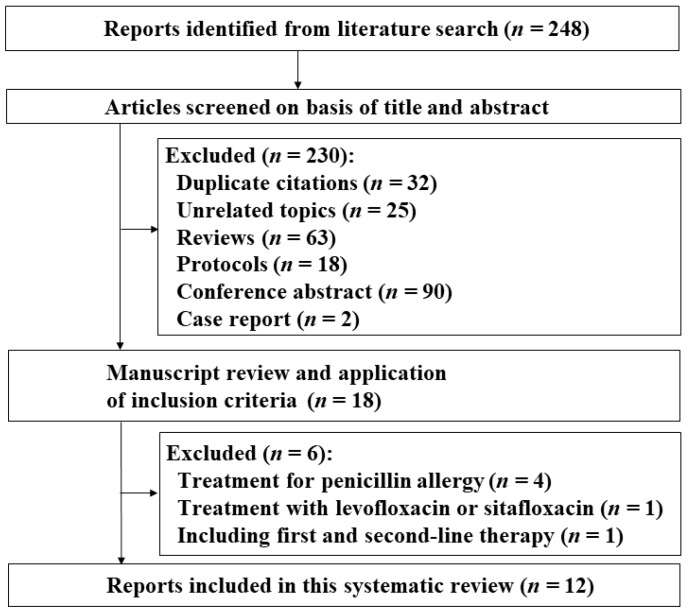
Flow diagram of the systematic literature search.

**Figure 2 jcm-10-02722-f002:**
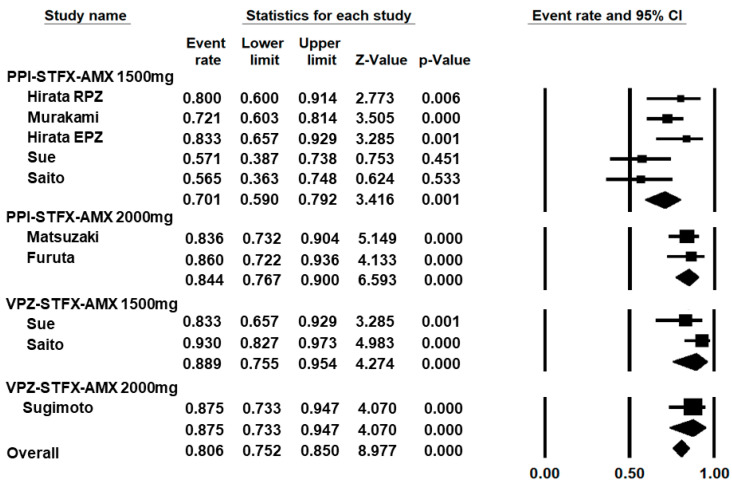
Eradication rates of 7 days regimen with PPI (proton pump inhibitor) or vonoprazan. (VPZ)–sitafloxacin (STFX)–amoxicillin (AMX). RPZ: rabeprazole; EPZ: esomeprazole.

**Figure 3 jcm-10-02722-f003:**
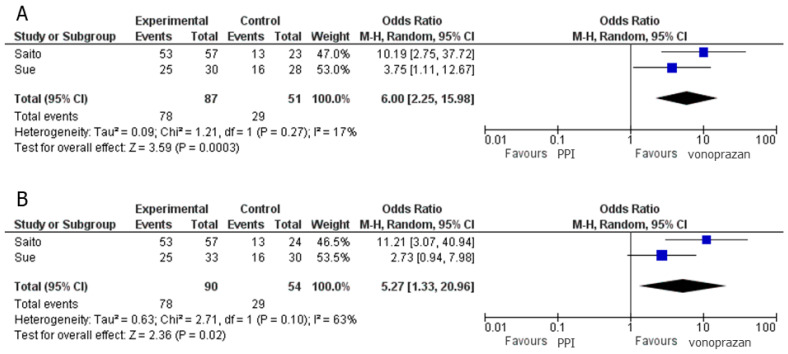
Forest plot of the pooled odds ratio (OR) with 95% confidence intervals (CI) of successful eradication when using regimens of vonoprazan–sitafloxacin–amoxicillin and PPI–sitafloxacin–amoxicillin. (**A**) Per-protocol analysis. (**B**) Intention-to-treat analysis.

**Figure 4 jcm-10-02722-f004:**
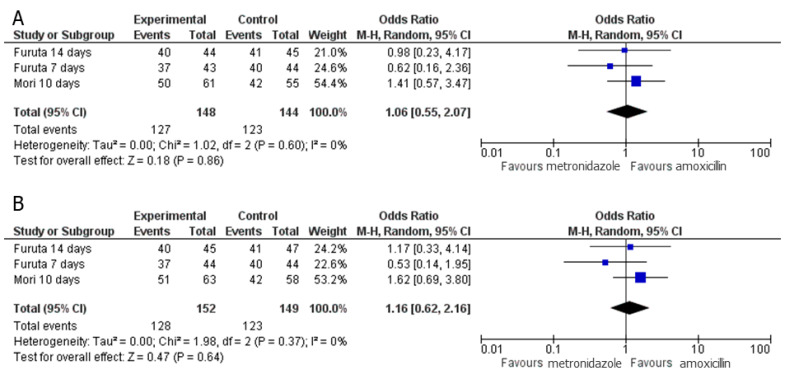
Forest plot of the pooled odds ratio (OR) with 95% confidence intervals (CI) of eradication failure when using regimens of PPI–sitafloxacin–amoxicillin and PPI–sitafloxacin–metronidazole. (**A**) Per-protocol analysis. (**B**) Intention-to-treat analysis.

**Figure 5 jcm-10-02722-f005:**
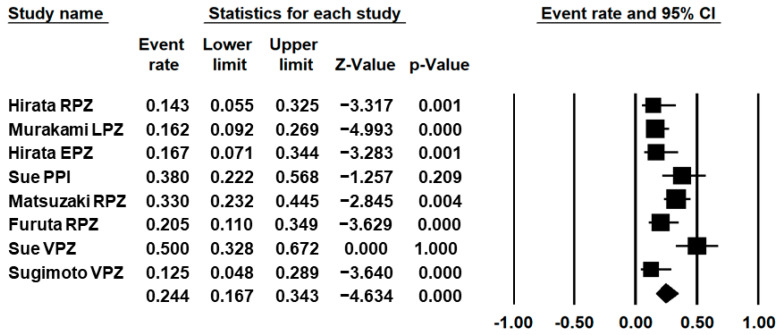
Adverse event (diarrhoea) of 7 days regimen with PPI or vonoprazan (VPZ)–sitafloxacin (STFX)–amoxicillin (AMX). Murakami et al.’s study included other adverse events. RPZ: rabeprazole; LPZ lansoprazole; EPZ: esomeprazole.

**Table 1 jcm-10-02722-t001:** Efficacy of PPI–sitafloxacin (STFX)–amoxicillin (AMX).

**Author**	**Year**	**Drug Combination**			**Days**	**Eradication Rate**		**STFX**
		**PPI**	**STFX**	**AMX**		**ITT**	**PP**	**Resistant Rate**
Hirata	2012	RPZ 20 m	200 mg	1500 mg	7	75.0	80.0	25.0
Murakami	2013	LPZ 60 mg	200 mg	1500 mg	7	70.0	72.1	―
Hirata	2016	EPZ 40 mg	200 mg	1500 mg	7	83.3	83.3	50.0
Sue	2019	PPI	200 mg	1500 mg	7	53.3	57.1	―
Saito	2019	PPI	200 mg	1500 mg	7	54.2	56.5	21.7
Matsuzaki	2012	RPZ 40 mg	200 mg	2000 mg	7	78.2	83.6	42.5
Furuta	2014	RPZ 20–40 mg	200 mg	2000 mg	7	84.1	86.4	―
Tokunaga	2014	RPZ 30 mg	200 mg	1500 mg	10	―	87.1	―
Mori	2016	EPZ 40 mg	200 mg	2000 mg	10	81.0	82.0	60.3
Mori	2020	EPZ 40 mg	200 mg	2000 mg	10	81.6	81.6	50.0
Furuta	2014	RPZ 20–40 mg	200 mg	2000 mg	14	88.9	90.9	―

Sitafloxacin resistant rate: minimum inhibitory concentration ≥ 0.12 μg/mL. PPI: proton pump inhibitor; RPZ: rabeprazole; LPZ: lansoprazole; EPZ: esomeprazole; ITT: intention-to-treat; PP: per-protocol.

**Table 2 jcm-10-02722-t002:** Efficacy of vonoprazan (VPZ)–sitafloxacin (STFX)–amoxicillin (AMX).

Author	Year	Drug Combination			Days	Eradication Rate		STFX
		VPZ	STFX	AMX		ITT	PP	Resistant Rate
Sue	2016	VPZ 40 mg	200 mg	1500 mg	7	75.8	83.3	―
Saito	2019	VPZ 40 mg	200 mg	1500 mg	7	93.0	93.0	30.0
Sugimoto	2020	VPZ 40 mg	200 mg	2000 mg	7	―	87.5	―

Sitafloxacin resistant rate: minimum inhibitory concentration ≥ 0.12 μg/mL.

**Table 3 jcm-10-02722-t003:** Efficacy of PPI–sitafloxacin (STFX)–metronidazole (MNZ).

Author	Year	Drug Combination			Days	Eradication Rate		STFX
		PPI	STFX	MNZ		ITT	PP	Resistant Rate
Furuta	2014	RPZ 20–40 mg	200 mg	500 mg	7	90.9	90.9	―
Sugimoto	2015	RPZ 40 mg	200 mg	500 mg	7	―	88.3	―
Mori	2016	EPZ 40 mg	200 mg	500 mg	10	72.4	76.4	56.9
Furuta	2014	RPZ 20–40 mg	200 mg	500 mg	14	87.2	91.1	―

Sitafloxacin resistant rate: minimum inhibitory concentration ≧ 0.12 μg/mL.

**Table 4 jcm-10-02722-t004:** Quality assessment according to type of study.

Author	Year	Type of Study	Quality Assessment	
			Jadad Scale *	QATCSS Score ^†^
Murakami	2013	Randomised controlled study: open label	3	―
Furuta	2014	Randomised controlled study: open label	3	―
Sue	2019	Randomised controlled study: open label	3	―
Matsuzaki	2012	Prospective single centre open label study	―	9
Hirata	2012	Prospective single center open label study	―	9
Tokunaga	2014	Single center open label study	―	6
Mori	2015	Prospective single center open label study	―	9
Sugimoto	2015	Single center open label study	―	8
Hirata	2016	Prospective multi-center open label study	―	9
Saito	2019	Single center open label study	―	8
Mori	2020	Prospective single center open label study	―	9
Sugimoto	2020	Single center open label study	―	8

*: Jadad scale reached a maximum score of 5. ^†^: Quality assessment tool for case series studies (QATCSS) reaches a maximum score of 9.

## Data Availability

No new data were created or analyzed in this study. Data sharing is not applicable to this article.

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
