# Peer review of "Sitafloxacin for Third-Line Helicobacter pylori Eradication: A Systematic Review"

_jcm, 2021, doi:10.3390/jcm10122722_

Round 1
Reviewer 1 Report
This is a review of sitofloxacin data from Japan. The potential advantage of this newer fluoroquinolone is that resistance to other fluoroquinolones does not affect the outcome so it is still effective despite levofloxacin resistance. However, the presence of specific sitofloxacin resistance leads to drug failure. There are conceptual and practical issues to this paper. The most important issue is that the regimen was never optimized. Quinolone therapy for H. pylori is dependent on the effectiveness of antisecretory therapy and the duration of therapy. The focused on the effectiveness of antisecretory therapy which proved to provide a minor increase in effectiveness but failed to consistently achieve acceptable cure rates (preferably 95% or greater but minimally acceptable at >90%). They also provide details about the proportion with sitofloxacin resistance in each study and the range of MIC’s. The pattern of resistance to other fluoroquinolones is of minor interest and is of no interest if it is the only information regarding resistance. As such the report is misleading.
Possibly, longer duration is not possible in Japan but nevertheless, the authors should describe data about fluoroquinolone regimes such as their dependency on duration and antisecretory drug effectiveness. There statement is not qualified whereas the studies were: Among 7 day therapies, etc. Of interest, vonoprazan becomes effective on day 1 whereas PPIs take 3 or more days to become effective such that one could visualize the data as 4 vs 7 full days of full antisecretory therapy. A number of qualification are needed to become acceptable and eliminate the comparative bias (i.e., if one had to choose among the therapies based only on the data supplied, vonoprazan-containing regimen appears marginally better. BUT, it is likely that higher cure rates can be obtained by increasing the duration of therapy and all the studies suffer from not having been optimized.
Finally, the US FDA and the European regulatory agency have recently added a number of warning regard use of fluoroquinolones and this needs to be added.
Author Response
Answer to Reviewer
Thank you for your important comments, which were extremely helpful for improving the quality of our manuscript.
This is a review of sitofloxacin data from Japan. The potential advantage of this newer fluoroquinolone is that resistance to other fluoroquinolones does not affect the outcome so it is still effective despite levofloxacin resistance. However, the presence of specific sitofloxacin resistance leads to drug failure. There are conceptual and practical issues to this paper. The most important issue is that the regimen was never optimized. Quinolone therapy for H. pylori is dependent on the effectiveness of antisecretory therapy and the duration of therapy. The focused on the effectiveness of antisecretory therapy which proved to provide a minor increase in effectiveness but failed to consistently achieve acceptable cure rates (preferably 95% or greater but minimally acceptable at >90%). They also provide details about the proportion with sitofloxacin resistance in each study and the range of MIC’s. The pattern of resistance to other fluoroquinolones is of minor interest and is of no interest if it is the only information regarding resistance. As such the report is misleading.
Thank you very much for your insightful comments. Yuan Y et al. also reported that increasing the duration of PPI-based triple therapy increases H. pylori eradication rates in their meta-analysis (Yuan et al: Optimum duration of regimens for Helicobacter pylori eradication. Cochrane Database Syst Rev 2013(12):CD008337). For PPI-clarithromycin-amoxicillin regimens, prolonging treatment duration from 7 to 10 or from 10 to 14 days is associated with a significantly higher eradication rate. Noh et al. also reported that increasing H. pylori
eradication rate by extending the duration of levofloxacin-based rescue therapy to 14 days (Korean J Gastroenterol 2016, 68(5):260-264). Therefore, extending the duration of vonoprazan-sitafloxacin-amoxicillin regimen is promising. The optimal duration of vonoprazan-sitafloxacin-amoxicillin regimen needs to be further investigated in future studies. These points were added into the revised Discussion section.
According to your comment, we deleted the pattern of resistance to levofloxacin in Table 1, and 2.
Possibly, longer duration is not possible in Japan but nevertheless, the authors should describe data about fluoroquinolone regimes such as their dependency on duration and antisecretory drug effectiveness. There statement is not qualified whereas the studies were: Among 7 day therapies, etc. Of interest, vonoprazan becomes effective on day 1 whereas PPIs take 3 or more days to become effective such that one could visualize the data as 4 vs 7 full days of full antisecretory therapy. A number of qualification are needed to become acceptable and eliminate the comparative bias (i.e., if one had to choose among the therapies based only on the data supplied, vonoprazan-containing regimen appears marginally better. BUT, it is likely that higher cure rates can be obtained by increasing the duration of therapy and all the studies suffer from not having been optimized.
As you pointed out, the optimal duration is important. Following sentences were added into the Discussion section.
“Regarding the optimal duration for H. pylori eradication therapy, a meta-analysis showed that increasing the duration of PPI-based triple therapy increased the eradication rates (Yuan Y et al, Cochrane Database Syst Rev 2013(12):CD008337). The eradication rate of levofloxacin-based rescue therapy was increased by extending the duration to 14 days (Noh et al. Korean J Gastroenterol. 2016,25;68(5):260-264). Conversely, Furura et al. compared 7-day and 14-days regimens of PPI-sitafloxacin-amoxicillin in their RCT, and found no significant difference. Mori et al. also compared 7-day and 10-days regimens of PPI-sitafloxacin-amoxicillin, and found no significant difference. As for the vonoprazan-sitafloxacin-amoxicillin regimen, there is no data other than 7-days regimen. The optimal duration of vonoprazan-sitafloxacin-amoxicillin regimen needs to be further investigated in future studies.”
Finally, the US FDA and the European regulatory agency have recently added a number of warning regard use of fluoroquinolones and this needs to be added.
Thank you very much for your valuable comment. Following sentences were added into the revised manuscript.
“United States Food and Drug Administration recently enhanced warnings about disabling and potentially permanent side effects involving the tendons, muscles, joints, nerves, and central nervous system (Buehrle DJ, et al. Antimicrob Agents Chemother 2021). Balancing risk and benefit should be considered.”
Reviewer 2 Report
Comments to the Author:
In this systematic review, the authors included 11 studies for meta-analysis to evaluate sitafloxacin for third-line H. pylori eradication. An appropriate salvage treatment for H. pylori infection after previous failures is important because of increasing prevalence of antibiotics resistant H. pylori infection. The study was well conducted. However, applicability of the results is limited because sitafloxacin-based regimen is only available in Japan (or Thailand). There are several point that need to be revised.
- Methods
Please explain why EMBASE databases were not searched.
- Methods
Please add explanations how the authors searched for gray literature.
- Methods and Results
The search term seems too simple, and there are a small number of studies initially identified. Please explain who prepared the search strategy, by librarian? or by clinician?
- Results
The efficacy of third line regimen for H. pylori eradication is affected by which regimens were failed previously. Thus, it is important to know which regimens were previously failed in understanding the efficacy of sitafloxacin based regimen and clinical applicability of this regimen. Please present the previously failed 1st line and 2nd lines regimens in each study.
- Table 3
Comprehensive Meta-Analysis (CMA) provides meta-analysis of single arm data. The mean eradication rates in the Table 3 can be replaced with pooled eradication rate using CMA. In addition, it also provides a subgroup analysis. Although the authors suggested that the eradication rate of PPI-sitafloxacin-amoxicillin regimen (1500mg/day) was inferior to those of PPI-sitafloxacin-amoxicillin (2000mg/day) and vonoprazan-sitafloxacin-amoxicillin (1500mg and 2000mg/day), this was not statistically evaluated. Please conduct subgroup analysis using CMA and describe the results based on the statistics.
- Page 4, lines 124 – 125.
“PPI-sitafloxacin-amoxicillin 2000 mg/day (88.9%; 95% CI, 75.5-95.4%), and vonoprazan-sitafloxacin-amoxicillin 1500 mg/day (87.5%; 95% CI, 73.2-95.8%).”
-> vonoprazan-sitafloxacin-amoxicillin 1500 mg/day (88.9%; 95% CI, 75.5-95.4%), and vonoprazan-sitafloxacin-amoxicillin 2000 mg/day (87.5%; 95% CI, 73.2-95.8%).”
- Figure 2
Please provide intention-to-treat (ITT) analysis results, too.
- Figure 3
Please provide intention-to-treat (ITT) analysis results, too.
- Table 5
Please provide pooled rate or adverse events (or diarrhea) using CMA.
- Discussion
There seem two additional limitations in this study. First, because this study included studies conducted in Japan only, the efficacy of sitafloxacin based regimen is affected by local antibiotics resistance profile. In addition, to my knowledge, in Japan, 1st line regimen is PPI-amoxicillin-clarithromycin and 2nd line regimen is PPI-amoxicillin-metronidazole. Therefore, the results of the current study may not directly applicable to the populations who were administered bismuth containing regimens as 1st line or 2nd line regimen. This point should be discussed in coupled with the previously failed regimens of the included studies as commented in comment #4. Second, the authors need to add limited availability of vonoprazan in addition to sitafloxacin.
Author Response
Answer to Reviewer
Thank you for your important comments, which were extremely helpful for improving the quality of our manuscript.
In this systematic review, the authors included 11 studies for meta-analysis to evaluate sitafloxacin for third-line H. pylori eradication. An appropriate salvage treatment for H. pylori infection after previous failures is important because of increasing prevalence of antibiotics resistant H. pylori infection. The study was well conducted. However, applicability of the results is limited because sitafloxacin-based regimen is only available in Japan (or Thailand). There are several point that need to be revised.
1.Methods
Please explain why EMBASE databases were not searched.
The use of EMBASE databases is expensive, and EMBASE databases are not available in our institutes. Most of the research papers for sitafloxacin are from Japan. Therefore, we used not only PubMed, and Cochrane library, but also Igaku-Chuo-Zasshi database, which is a Japanese database.
2.Methods
Please add explanations how the authors searched for gray literature.
As you pointed out, there are a small number of studies initially identified. So, search terms were modified from “third-line or rescue” to “eradication”. The systematic review process identified 173 potential reports (revised Figure 1). Articles were screened on basis of title and abstract, and we excluded 155 studies (10 duplications, 9 unrelated topics, 57 review articles, 10 protocols for clinical trials, 60 meeting abstracts, and 3 case reports) based on the exclusion criteria. The remaining 12 studies were examined in detail, and six additional studies were rejected. In four studies, the sitafloxacin-containing regimen was used as first-line or second-line treatment for subjects with penicillin allergy. One study combined the eradication rates of levofloxacin-based and sitafloxacin-based therapy. One study combined the eradication rates of first, second, and third-line therapy. Finally, we included 12 clinical studies in the present systematic review. This modified search found another article written in Japanese (Tokunaga et al. J germfree life gnotobiol 2014, 44(1):38-40.)
Although the data of this article were added into the revised manuscript, the main results did not change.
3.Methods and Results
The search term seems too simple, and there are a small number of studies initially identified. Please explain who prepared the search strategy, by librarian? or by clinician?
Two researchers (T.N. and M.M.) independently searched relevant reports. Two researchers are clinicians (MD., Ph.D.). The final search terms were decided through discussion. As you pointed out, there are a small number of studies initially identified. So, search terms were modified, and Figure 1 was also modified.
4.Results
The efficacy of third line regimen for H. pylori eradication is affected by which regimens were failed previously. Thus, it is important to know which regimens were previously failed in understanding the efficacy of sitafloxacin based regimen and clinical applicability of this regimen. Please present the previously failed 1st line and 2nd lines regimens in each study.
In Sugimoto et al.’s study, first-line therapy was the 7-day regimen with vonoprazan 20 mg b.i.d., clarithromycin 200 mg b.i.d., and amoxicillin 750 mg b.i.d., and second-line therapy was the 7-day regimen with vonoprazan 20 mg b.i.d., metronidazole 250 mg b.i.d., and amoxicillin 750 mg b.i.d. In Saito et al.’s and Sue et al.’ studies, first-line therapy was the 7-day regimen with vonoprazan or PPI, clarithromycin 200 mg or 400mg b.i.d., and amoxicillin 750 mg b.i.d., and second-line therapy was the 7-day regimen with vonoprazan or PPI, metronidazole 250 mg b.i.d., and amoxicillin 750 mg b.i.d. In other 8 studies, first-line therapy was the 7-day regimen with PPI, clarithromycin 200 mg or 400mg b.i.d., and amoxicillin 750 mg b.i.d., and second-line therapy was the 7-day regimen with PPI, metronidazole 250 mg b.i.d., and amoxicillin 750 mg b.i.d. These points were added into the revised manuscript.
5.Table 3
Comprehensive Meta-Analysis (CMA) provides meta-analysis of single arm data. The mean eradication rates in the Table 3 can be replaced with pooled eradication rate using CMA. In addition, it also provides a subgroup analysis. Although the authors suggested that the eradication rate of PPI-sitafloxacin-amoxicillin regimen (1500mg/day) was inferior to those of PPI-sitafloxacin-amoxicillin (2000mg/day) and vonoprazan-sitafloxacin-amoxicillin (1500mg and 2000mg/day), this was not statistically evaluated. Please conduct subgroup analysis using CMA and describe the results based on the statistics.
According to your comment, Table 3 was replaced with the forest plot of eradication rate using Comprehensive Meta-Analysis.
For the PPI-sitafloxacin-amoxicillin regimen (1500 mg/day), the mean eradication rate was 70.1% (95% CI, 59.0-79.2%). Although this was tended to be inferior when compared with PPI-sitafloxacin-amoxicillin 2000 mg/day (84.4%; 95% CI, 76.7-90.0%), vonoprazan-sitafloxacin-amoxicillin 1500 mg/day (88.9%; 95% CI, 75.5-95.4%), and vonoprazan-sitafloxacin-amoxicillin 2000 mg/day (87.5%; 95% CI, 73.3-94.7%), the difference was not statistically significant. Therefore, the sentence of “this was found to be inferior” was modified to “this was tended to be inferior” in the Result and Discussion section.
6.Page 4, lines 124 – 125.
“PPI-sitafloxacin-amoxicillin 2000 mg/day (88.9%; 95% CI, 75.5-95.4%), and vonoprazan-sitafloxacin-amoxicillin 1500 mg/day (87.5%; 95% CI, 73.2-95.8%).”
-> vonoprazan-sitafloxacin-amoxicillin 1500 mg/day (88.9%; 95% CI, 75.5-95.4%), and vonoprazan-sitafloxacin-amoxicillin 2000 mg/day (87.5%; 95% CI, 73.2-95.8%).”
Thank you very much. It was corrected.
7.Figure 2
Please provide intention-to-treat (ITT) analysis results, too.
According to your comment, we added intention-to-treat (ITT) analysis results in revised Figure 3 (previous Figure 2). The intention-to-treat analysis had similar results (Figure 3B).
8.Figure 3
Please provide intention-to-treat (ITT) analysis results, too.
According to your comment, we added intention-to-treat (ITT) analysis results in revised Figure 4 (previous Figure 3). The intention-to-treat analysis had similar results (Figure 4B).
9.Table 5
Please provide pooled rate or adverse events (or diarrhea) using CMA.
Table 5 was replaced with the forest plot of adverse event (diarrhoea) rate using Comprehensive Meta-Analysis (Figure 5). The pooled adverse event (diarrhoea) rate was 24.4% (95% CI: 16.7-34.3%).
10.Discussion
There seem two additional limitations in this study. First, because this study included studies conducted in Japan only, the efficacy of sitafloxacin based regimen is affected by local antibiotics resistance profile. In addition, to my knowledge, in Japan, 1st line regimen is PPI-amoxicillin-clarithromycin and 2nd line regimen is PPI-amoxicillin-metronidazole. Therefore, the results of the current study may not directly applicable to the populations who were administered bismuth containing regimens as 1st line or 2nd line regimen. This point should be discussed in coupled with the previously failed regimens of the included studies as commented in comment #4. Second, the authors need to add limited availability of vonoprazan in addition to sitafloxacin.
Thank you very much for your important comments. In Japan, first-line regimen is PPI-clarithromycin- amoxicillin and second-line regimen is PPI-metronidazole- amoxicillin. Therefore, the results of the current study may not directly applicable to the populations who were administered bismuth containing regimens as first-line or second-line regimen. Therefore, these results may not be generalisable to other ethnicities and countries. The availability of vonoprazan is also limited in addition to sitafloxacin. The points were added into the revised manuscript (the limitation section of Discussion).
Round 2
Reviewer 1 Report
Their conclusion is suspect in that recent data suggests that vonoprazan-amoxicillin dual therapy is likely superior. 1, 2 In addition, the data about 7 vs. 14 day fluoroquinolone therapy is larger than one reference 3-8. At best they could say it is an option but should recommend 14-day therapy with the caveat about potential dangers. Susceptibility-based therapy is the best option
- Furuta T, Yamade M, Kagami T, et al. Dual Therapy with Vonoprazan and Amoxicillin Is as Effective as Triple Therapy with Vonoprazan, Amoxicillin and Clarithromycin for Eradication of Helicobacter pylori. Digestion 2019:1-9.
- Suzuki S, Gotoda T, Kusano C, et al. Seven-day vonoprazan and low-dose amoxicillin dual therapy as first-line Helicobacter pylori treatment: a multicentre randomised trial in Japan. Gut 2020;69:1019-1026.
- Noh HM, Hong SJ, Han JP, et al. Eradication Rate by Duration of Third-line Rescue Therapy with Levofloxacin after Helicobacter pylori Treatment Failure in Clinical Practice. The Korean journal of gastroenterology = Taehan Sohwagi Hakhoe chi 2016;68:260-264.
- Ercin CN, Uygun A, Toros AB, et al. Comparison of 7- and 14-day first-line therapies including levofloxacin in patients with Helicobacter pylori positive non-ulcer dyspepsia. Turk. J Gastroenterol 2010;21:12-16.
- Miehlke S, Krasz S, Schneider-Brachert W, et al. Randomized trial on 14 versus 7 days of esomeprazole, moxifloxacin, and amoxicillin for second-line or rescue treatment of Helicobacter pylori Iinfection. Helicobacter 2011;16:420-426.
- Ozdil K, Calhan T, Sahin A, et al. Levofloxacin based sequential and triple therapy compared with standard plus probiotic combination for Helicobacter pylori eradication. Hepatogastroenterology 2011;58:1148-1152.
- Chuah SK, Tai WC, Hsu PI, et al. The efficacy of second-line anti-Helicobacter pylori therapy using an extended 14-day levofloxacin/amoxicillin/proton-pump inhibitor treatment--a pilot study. Helicobacter 2012;17:374-381.
- Liao J, Zheng Q, Liang X, et al. Effect of fluoroquinolone resistance on 14-day levofloxacin triple and triple plus bismuth quadruple therapy. Helicobacter 2013;18:373-377.
Author Response
Answer to Reviewer
Thank you for your important comments, which were extremely helpful for improving the quality of our manuscript.
Their conclusion is suspect in that recent data suggests that vonoprazan-amoxicillin dual therapy is likely superior. 1, 2 In addition, the data about 7 vs. 14 day fluoroquinolone therapy is larger than one reference 3-8. At best they could say it is an option but should recommend 14-day therapy with the caveat about potential dangers. Susceptibility-based therapy is the best option
Furuta T, Yamade M, Kagami T, et al. Dual Therapy with Vonoprazan and Amoxicillin Is as Effective as Triple Therapy with Vonoprazan, Amoxicillin and Clarithromycin for Eradication of Helicobacter pylori. Digestion 2019:1-9.
Suzuki S, Gotoda T, Kusano C, et al. Seven-day vonoprazan and low-dose amoxicillin dual therapy as first-line Helicobacter pylori treatment: a multicentre randomised trial in Japan. Gut 2020;69:1019-1026.
Noh HM, Hong SJ, Han JP, et al. Eradication Rate by Duration of Third-line Rescue Therapy with Levofloxacin after Helicobacter pylori Treatment Failure in Clinical Practice. The Korean journal of gastroenterology = Taehan Sohwagi Hakhoe chi 2016;68:260-264.
Ercin CN, Uygun A, Toros AB, et al. Comparison of 7- and 14-day first-line therapies including levofloxacin in patients with Helicobacter pylori positive non-ulcer dyspepsia. Turk. J Gastroenterol 2010;21:12-16.
Miehlke S, Krasz S, Schneider-Brachert W, et al. Randomized trial on 14 versus 7 days of esomeprazole, moxifloxacin, and amoxicillin for second-line or rescue treatment of Helicobacter pylori Iinfection. Helicobacter 2011;16:420-426.
Ozdil K, Calhan T, Sahin A, et al. Levofloxacin based sequential and triple therapy compared with standard plus probiotic combination for Helicobacter pylori eradication. Hepatogastroenterology 2011;58:1148-1152.
Chuah SK, Tai WC, Hsu PI, et al. The efficacy of second-line anti-Helicobacter pylori therapy using an extended 14-day levofloxacin/amoxicillin/proton-pump inhibitor treatment--a pilot study. Helicobacter 2012;17:374-381.
Liao J, Zheng Q, Liang X, et al. Effect of fluoroquinolone resistance on 14-day levofloxacin triple and triple plus bismuth quadruple therapy. Helicobacter 2013;18:373-377.
Thank you very much. Inspired by your valuable comments, we revised into the innovative paper.
Following sentences were added into the revised Discussion section.
“Regarding the optimal duration for H. pylori eradication therapy, a meta-analysis showed that increasing the duration of PPI-based triple therapy increased the eradication rates [45]. There are many reports that the eradication rates of fluoroquinolone-based therapy were increased by extending the duration to 14 days [46-52]. Conversely, Furura et al. compared 7-day and 14-days regimens of PPI-sitafloxacin-amoxicillin in their RCT, and found no significant difference. Mori et al. also compared 7-day and 10-days regimens of PPI-sitafloxacin-amoxicillin, and found no significant difference. However, the number of patients was limited, and hence these studies might be underpowered. As for the vonoprazan-sitafloxacin-amoxicillin regimen, there is no data other than 7-days regimen. The 14-day regimen with vonoprazan-sitafloxacin-amoxicillin may achieve excellent eradication rate. The optimal duration of vonoprazan-sitafloxacin-amoxicillin regimen needs to be further investigated in future studies.”
[45] Yuan Y, Ford AC, Khan KJ, Gisbert JP, Forman D, Leontiadis GI, Tse F, Calvet X, Fallone C, Fischbach L et al: Optimum duration of regimens for Helicobacter pylori eradication. Cochrane Database Syst Rev 2013(12):CD008337.
[46] Noh HM, Hong SJ, Han JP, Park KW, Lee YN, Lee TH, Ko BM, Lee JS, Lee MS: Eradication Rate by Duration of Third-line Rescue Therapy with Levofloxacin after Helicobacter pylori Treatment Failure in Clinical Practice. Korean J Gastroenterol 2016, 68(5):260-264.
[47] Ercin CN, Uygun A, Toros AB, Kantarcioglu M, Kilciler G, Polat Z, Bagci S: Comparison of 7- and 14-day first-line therapies including levofloxacin in patients with Helicobacter pylori positive non-ulcer dyspepsia. Turk J Gastroenterol 2010, 21(1):12-16.
[48] Miehlke S, Krasz S, Schneider-Brachert W, Kuhlisch E, Berning M, Madisch A, Laass MW, Neumeyer M, Jebens C, Zekorn C et al: Randomized trial on 14 versus 7 days of esomeprazole, moxifloxacin, and amoxicillin for second-line or rescue treatment of Helicobacter pylori infection. Helicobacter 2011, 16(6):420-426.
[49] Li BZ, Threapleton DE, Wang JY, Xu JM, Yuan JQ, Zhang C, Li P, Ye QL, Guo B, Mao C et al: Comparative effectiveness and tolerance of treatments for Helicobacter pylori: systematic review and network meta-analysis. BMJ 2015, 351:h4052.
[50] Chuah SK, Tai WC, Hsu PI, Wu DC, Wu KL, Kuo CM, Chiu YC, Hu ML, Chou YP, Kuo YH et al: The efficacy of second-line anti-Helicobacter pylori therapy using an extended 14-day levofloxacin/amoxicillin/proton-pump inhibitor treatment--a pilot study. Helicobacter 2012, 17(5):374-381.
[51] Liao J, Zheng Q, Liang X, Zhang W, Sun Q, Liu W, Xiao S, Graham DY, Lu H: Effect of fluoroquinolone resistance on 14-day levofloxacin triple and triple plus bismuth quadruple therapy. Helicobacter 2013, 18(5):373-377.
[52] Ozdil K, Calhan T, Sahin A, Senates E, Kahraman R, Yuzbasioglu B, Demirdag H, Demirsoy H, Sokmen MH: Levofloxacin based sequential and triple therapy compared with standard plus probiotic combination for Helicobacter pylori eradication. Hepatogastroenterology 2011, 58(109):1148-1152.
The references of No. 45-52 were added in the revising process.
Regarding the drug susceptibility, following sentence was added into the revised Discussion section.
“Eradication rates for gyrA mutation-negative and mutation-positive groups were 96.7% and 74.4% in Matsuzaki et al.'s study [21], 100% and 70.3% in Mori et al.'s 2016 study [24], and 89.5% and 68.4% in Mori et al.'s 2020 study, respectively [25]. Sitafloxacin-based triple therapy showed excellent eradication rates for gyrA mutation-negative H. pylori. Sitafloxacin-based therapy should be used for sitafloxacin susceptible or gyrA mutation-negative H. pylori. Sitafloxacin resistant or gyrA mutation-positive H. pylori would require alternate treatments such as rifabutin-based therapy [13, 59-61].”
Furthermore, the conclusion was revised as follows.
“In conclusion, the 7-day regimen with vonoprazan 20 mg b.i.d., sitafloxacin 100 mg b.i.d., and amoxicillin 750 mg b.i.d. or 500 mg q.i.d. is a good option as the third-line H. pylori eradication treatment, at least in Japan. However, the extension of treatment duration should be considered to further improve the eradication rate. Considering the safety concern of fluoroquinolones, sitafloxcin should be used after confirming drug susceptibility.”
Reviewer 2 Report
Comments to the Author:
I appreciate the authors’ efforts in the revision process. However, there are several essential points that should be revised.
- Methods.
The authors ‘must’ include the results of EMBASE database search. It is one of core databases.
- Methods
The search strategy should be updated. For example, search terms for H. pylori infection may include ‘helicobacter pylori’ and ‘helicobacter infection’. They also may include proximity operators like ‘near’. It is recommended that the authors consult to experienced librarian for this issue. The current version of search is more like a review rather than a ‘systematic’ review.
- Results, page 4, lines 125 – 131.
“For the PPI-sitafloxacin-amoxicillin regimen (1500 mg/day), the mean eradication rate was 70.1% (95% CI, 59.0-79.2%); however, this was tended to be inferior when compared with…”
Please removed the expression ‘tended to be inferior’ because it is not an objective description. Please present the P-value of the subgroup analysis and describe that the difference was not statistically significant.
Author Response
Answer to Reviewer
Thank you for your important comments, which were extremely helpful for improving the quality of our manuscript.
I appreciate the authors’ efforts in the revision process. However, there are several essential points that should be revised.
Methods.
The authors ‘must’ include the results of EMBASE database search. It is one of core databases.
I am sorry. The usage fee for EMBASE is high, and our institutes (International University of Health and Welfare, and Tokai University) do not have a contract with EMBASE. Therefore, we cannot use it. Instead of EMBASE, we added Web of Science for the literature search. We believe that we detected the necessary articles, because we repeatedly searched using PubMed, Cochrane library, Web of Science, and Igaku-Chuo-Zasshi database in Japan with various search words.
Methods
The search strategy should be updated. For example, search terms for H. pylori infection may include ‘helicobacter pylori’ and ‘helicobacter infection’. They also may include proximity operators like ‘near’. It is recommended that the authors consult to experienced librarian for this issue. The current version of search is more like a review rather than a ‘systematic’ review.
Thank you very much for your professional comments.
In an article of “Rifabutin triple therapy for first-line and rescue treatment of Helicobacter pylori infection: A systematic review and meta-analysis”, the search terms were “Helicobacter pylori” OR “H. pylori” AND “amoxicillin” AND “rifabutin” (Gingold-Belfer et al. J Gastroenterol Hepatol. 2020, PMID: 33037845). In an article of “Systematic review with meta-analysis: the efficacy of levofloxacin triple therapy as the first- or second-line treatments of Helicobacter pylori infection”, the search terms were “levofloxacin” and “triple therapy” or “levofloxacin” and “Helicobacter pylori” or “Helicobacter” or “H. pylori” (Che et al. Aliment Pharmacol Ther. 2016 Sep;44(5):427-37). In our paper, we modified the search terms from “Helicobacter pylori” AND “sitafloxacin” AND “eradication” to (“Helicobacter pylori” OR “H. pylori” OR “Helicobacter infection”) AND (“sitafloxacin” OR “DU-6859a” OR “Gracevit”) AND (“eradication” OR “treatment” OR “therapy”). The number of potential reports increased from 173 to 248. The search results have been changed as follows.
“The systematic review process identified 248 potential reports (Figure 1). Based on the exclusion criteria, we then excluded 230 studies (32 duplications, 25 unrelated topics, 63 review articles, 18 protocols for clinical trials, 90 meeting abstracts, and 2 case reports). The remaining 18 studies were scrutinised, and six additional studies were rejected.”
Figure 1 was also modified.
Results, page 4, lines 125 – 131.
“For the PPI-sitafloxacin-amoxicillin regimen (1500 mg/day), the mean eradication rate was 70.1% (95% CI, 59.0-79.2%); however, this was tended to be inferior when compared with…”
Please removed the expression ‘tended to be inferior’ because it is not an objective description. Please present the P-value of the subgroup analysis and describe that the difference was not statistically significant.
According to your comment, we removed the expression ‘tended to be inferior’. We considered that it was not statistically significant because the 95% confidence intervals partially overlapped. It could be biased to compare the eradication rates between studies with different backgrounds, so the revised manuscript does not present the p-values. We described the mean eradication rates with 95% confidence intervals in overall analysis and subgroup analyses. The description of Result section has been changed as follows.
“Figure 2 shows the forest plot of eradication rates for 7-day regimens of either PPI or vonoprazan-sitafloxacin-amoxicillin. The mean eradication rate for 7-day regimens of either PPI or vonoprazan-sitafloxacin-amoxicillin was 80.6% (95% CI, 75.2-85.0). In the subgroup analyses, the mean eradication rates were 70.1% (95% CI, 59.0-79.2%) for PPI-sitafloxacin-amoxicillin 1500 mg/day, 84.4% (95% CI, 76.7-90.0%) for PPI-sitafloxacin-amoxicillin 2000 mg/day, 88.9% (95% CI, 75.5-95.4%) for vonoprazan-sitafloxacin-amoxicillin 1500 mg/day, and 87.5% (95% CI, 73.3-94.7%) vonoprazan-sitafloxacin-amoxicillin 2000 mg/day).”
Figure 2 was also modified to add the overall analysis.
In the Discussion section, the discussion about the optimal amoxicillin dose was modified as follows.
“In PPI-sitafloxacin-amoxicillin, amoxicillin is administered with 1500 mg/day (750 mg b.i.d.), or 2000 mg/day (500 mg q.i.d.). The bactericidal effect of amoxicillin depends on the time above the minimum inhibitory concentration (MIC), as amoxicillin has minimal post-antibiotic effect [42, 43]. Furuta et al. compared amoxicillin 750 mg b.i.d., 500 mg t.i.d., and 500 mg q.i.d. in a PPI-metronidazole-amoxicillin regimen, and reported eradication rates of 80.5%, 90.5%, and 95.2%, respectively, indicating that four times daily amoxicillin dosing maximised the eradication rate [44]. This study indicated that the optimal amoxicillin dose was 2000 mg/day (500 mg q.i.d.). The amoxicillin dose in the vonoprazan-sitafloxacin-amoxicillin regimen needs to be further investigated in future studies.